# Geographical accessibility to the supply of antiophidic sera in Brazil: Timely access possibilities

**Ricardo Antunes Dantas de Oliveira**[1]*, **Diego Ricardo Xavier Silva**[1], **Maurício Gonçalves e Silva**[2]

1 Health Information Laboratory, Institute of Communication and Scientific and Technological Information in Health (Icict), Oswaldo Cruz Foundation (Fiocruz), Rio de Janeiro (RJ), Brazil, 2 Brazilian Institute of Geography and Statistics (IBGE), Rio de Janeiro (RJ), Brazil

☯ These authors contributed equally to this work.

* ricardo.dantas@icict.fiocruz.br

**Data Availability Statement:** The datasets and basic map generated during the study development is available with no restriction at https://doi.org/10.7303/syn26042133. The Snakebite accidents data that support the findings of this study are available

## Abstract

Snakebite accidents are considered category A neglected tropical diseases. Brazil stands out for snakebite accidents, mainly in the Amazon region. The best possible care after snakebite accidents is to obtain antiophidic sera on time. And the maximum ideal time to reach it is about 2 hours after an accident. Based on public health information and using a tool to analyze geographical accessibility, we evaluate the possibility of reaching Brazilian serum-providing health facilities from the relationship between population distribution and commuting time. In this exploratory descriptive study, the geographic accessibility of Brazilian population to health facilities that supply antiophidic serum is evaluated through a methodology that articulates several issues that influence the commuting time to health units (ACCESSMOD): population and facilities' distribution, transportation network and means, relief and land use, which were obtained in Brazilian and international sources. The relative importance of the population without the possibility of reaching a facility in two hours is highlighted for Macro-Regions, States and municipalities. About nine million people live in locations more than two hours away from serum-providing facilities, with relevant variations between regions, states, and municipalities. States like Mato Grosso, Pará and Maranhão had the most important participation of population with reaching time problems to those units. The most significant gaps are found in areas with a dispersed population and sometimes characterized by a high incidence of snakebites, such as in the North of the country, especially in the Northeastern Pará state. Even using a 2010 population distribution information, because of the 2020 Census postponement, the tendencies and characteristics analyzed reveal challenging situations over the country. The growing availability of serum-providing health facilities, the enhanced possibilities of transporting accident victims, and even the availability of sera in other types of establishments are actions that would allow expanding the possibilities of access to serum supply.

from SINAN - Notifiable Diseases Information System, at http://www2.datasus.gov.br/DATASUS/index.php?area=0203&id=29878153. The Statistical Grid data that support the findings of this study are available in IBGE - Brazilian Institute of Geography and Statistics, at https://portaldemapas.ibge.gov.br/portal.php#homepage. The road network and land use data that support the findings of this study are available from OPEN STREET MAP (OSM), at: https://www.openstreetmap.org/. The Digital Elevation Model data that support the findings of this study are available from Earth Data - Open Access for Open Science, at: https://earthdata.nasa.gov/eosdis/daacs/lpdaac. The health facilities data that support the findings of this study are available from CNES - National Register of Health Facilities, at http://www2.datasus.gov.br/DATASUS/index.php?area=0204&id=6906 The 2019 population estimates data that support the findings of this study are available in IBGE - Brazilian Institute of Geography and Statistics, at https://www.ibge.gov.br/estatisticas/sociais/populacao/9103-estimativas-de-populacao.html?=&t=o-que-e.

**Funding:** RAD Oliveira have funding from Inova Fiocruz Program (https://portal.fiocruz.br/programa-inova-fiocruz), (Grant number: Inova Program VPPCB-008-FIO-18-2-48), a research support obtained in a selection occurred in 2018. The funds allowed the payment of scholarships to organize the datasets and the spatial information treatment, beyond the participation in congresses and seminars. The funding body had no role in the study design, data collection and analysis, decision to publish, or preparation of the manuscript.

**Competing interests:** The authors have declared that no competing interests exist.

## Introduction

In 2017, the World Health Organization classified snakebite poisoning as category A neglected tropical diseases [1], which spurred studies on appropriate prevention, implementable interventions, and resources to be allocated nationally and regionally [2].

Areas such as the Western Amazon, Sub-Saharan Africa, Southeast Asia, and Eastern Australia have many snake species. However, only some countries have effective sera for specific treatment by poisoning type [3]. In many cases, people use traditional therapies and treatments instead of the medical network and health centers, which leads to higher mortality and amputation rates. The predominance of such behavior is highlighted in African countries [2].

The morphoclimatic features of the Brazilian territory is related to the presence of snakes in all its Biomes. The profile of the population most affected by snakebite accidents has remained stable throughout the twentieth century, consisting of male rural workers aged 15–49 years [4].

Brazil registered a mean of 27,000 records of snakebites from 2001 to 2012 [5], with similar values until 2019, based on Notifiable Diseases Information System (SINAN) information [6]. Also, according to this system, the highest number of cases in recent years occurred in 2019, with 30,482, and the lowest in 2014, with 26,145.

The absolute majority, above 70% of the snakebite cases, were caused by the genus *Bothrops*, better known as Jararaca, followed by the genus *Crotalus* (7.5%), popularly identified as Cascavel. It should be noted that 11.5% of the cases failed to register information about the snake genus in SINAN [6] during the 2007–2019 period.

Also, according to SINAN [6], the regional distribution of the number of snake accidents is quite different when considering the totals from 2007 to 2019. The North region recorded 32% of the 367,199 accidents in the period, followed by the Northeast and Southeast, with 25.7% and 23.8%, respectively. Minor shares were registered in the Midwest (10.2%) and South (9%) regions.

Geographical accessibility is part of the broader, diverse and complex concept of access to health care [7, 8]. The evaluation of the quality of access to care involves not only geographical accessibility to health services, but also the availability of the them, its viability to obtain and the acceptability of care [9–11]. These various dimensions and its sub dimensions have multiple interactions and impacts on each other [9]. Specifically, geographical accessibility is related to the spatial distribution of health facilities and the time to access them, considering the travel costs related [10].

As many other health care types, the time to access health facilities and travel costs hinder access [10–12] to the supply of antiophidic sera. The geographical accessibility to modern and appropriate treatments in snake accidents is essential to ensure care [3]. A resolutive health unit two hours away increases the likelihood of successful treatment [13].

While snakebite poisoning is recognized as a neglected disease [1], significant lacunas are identified in the recent literature regarding geographic accessibility to health facilities that offer antiophidic sera. Few materials have been found in Brazil and international literature. Snakebite accidents and therapeutic care were recently analyzed in municipalities like Vassouras (RJ) [14] and Cruzeiro do Sul (AC), in the Brazilian Amazon [15]. Although they did not directly highlight geographic accessibility, they addressed the conditions for obtaining treatment in the respective municipalities. These analyses were performed on a local scale, distinct from our approach.

An example from international literature is the analysis of populations with difficult access to the supply of sera in Costa Rica, considering the distribution of poisonous snake species, the incidence of accidents, and commuting time to access health facilities [13]. A geoprocessing

software articulated topography information and transport routes and modes to calculate the time for accessing services.

Elements such as the spatial distribution of the population, the road network, the location of health centers and hospitals, and commuting time must be considered for a proper identification of the possible access to facilities providing antiophidic sera. Thus, the study aims to analyze the geographical accessibility to Brazilian serum-providing health facilities from the relationship between population distribution and commuting time.

## Methods

This exploratory descriptive study was developed with the use of public health statistics and other information, through ACCESSMOD 5 [16], a tool made available by World Health Organization to analyze the time/distance relationship in accessing health services. Based on travel time cost surface models, this software integrates information from different dimensions that influence commuting time, such as relief, routes and means of transportation, land use and population distribution [17, 18]. It differs from other methods because it considers the population based on a statistical grid instead of municipal centroids and distances from the different commuting types and their respective average speeds, which is not employed in most of the literature on the issue [17].

Information from different sources was used to analyze geographical accessibility to serum supply. The sources and preparation of each data to meet the study's needs are described below. Noteworthy is that all data have been redesigned for the Albers Equivalent Projection, with the same parameters used in the preparation of the IBGE statistical grid [19] (Central Meridian: -54˚; Latitude of Origin: -12˚; 1st Standard Parallel: -2˚; 2nd Standard Parallel: -22˚; Origin E: 5.000.000, Origin N: 10,000,000). Another common fact is that all data processing procedures were performed in Quantum GIS 3.4.1 and its extensions.

### Population

The population data derive from the IBGE statistical grid (IBGE, 2016) [19] and refer to 2010 since it was based on the last Demographic Census in Brazil. The base initially has a resolution of 200 meters for urban areas and 1 kilometer for other areas. A 200-meter resolution grid was created for Brazil as whole, perfectly superimposed on the statistical grid for use in the study. Centroids were generated for all of these features (squares). Then, these centroids were gathered spatially with the statistical grid to incorporate population and identification information into them. Subsequently, the population data of the centroids that fell into squares in the one-kilometer statistical grid was divided by 25 (5 rows x 5 columns) since that the number of centroids fits in a grid of this size. With a resolution of 200 meters and the 2010 population supplied, the resulting centroid grid was converted to the raster format.

### Routes

The routes used in the study are part of the 2018 Open Street Map (OSM) database [20], basically from two files: gis.osm_roads_free_1 and gis.osm_waterways_free_1, both in shapefile format. The classes of land routes were used in the first, and waterways were considered in the second. The features of this last file have been filtered to employ only those internal to the states of the North region, except for Tocantins, since, in many cases, they are the main and only access routes in that part of the country. The resulting two files were merged and kept in the shapefile format. Mean travel speeds were also assigned according to the classes of the OSM files, ranging from a maximum of 80 km/h on main roads to a minimum of 5 km/h on pedestrian routes, considering 20 km/h for commuting in rivers and channels.

## Land use

As in the route data, land use also derives from the OSM database [20], file gis.osm_landu-se_a_free_1. The original data did not change significantly since it was only redesigned and converted to the raster format. The ".img" format and "code" target field (that is, the land-use class code) were used as parameters to create the file. The speed considered for all classes was 1 km/h.

## Digital Elevation Model (DEM)

The influence of the relief on the time to reach a facility with an offer of antiophidic sera was considered in this study. Data from NASA's SRTM (Shuttle Radar Topography Mission) [21], version 3, with three seconds of arc size, were used to generate the DEM tracking, which is equivalent to 90 ground meters on the Equator. The original data were merged and resampled to 200 meters to match the other themes used in the study.

## Facilities

The health facilities offering antiophidic sera were provided by the Toxic-Pharmacological Information System (SINITOX) team [22], based on the National Register of Health Establishments (CNES) database [23, 24] and later georeferenced with automatic and manual methods, which in some cases demanded visual confirmation through Google Maps [25] and Google Street View [26], spatially concentrated in the state of Pará. We managed to position 2,198 units with capacity to provide this care type in Brazil at the end of the georeferencing process of health units. We had to consider the location coordinates of the local area where 25 facilities had their registered address since it was impossible to obtain their exact location.

## Distance and time to reach health facilities calculation

Population commuting time to reach health facilities was calculated using integrated information on population distribution, routes, land use, digital elevation model, and health facilities. The procedure was performed using the ACCESMOD 5 software [16], freely disseminated by the World Health Organization, that provides the simulation of the traveling time to the nearest health facility considered, using a travel time cost surface model. After the upload of the raster files of all the elements described above, the software generates catchment areas of health facilities using the Dijkstra least-cost path algorithm [17, 27].

The use of ACCESSMOD [16] allows further refinement when considering several dimensions for calculating the relationship between population and commuting time since it considers the relief, the speed by route type, and land use. As a result, the software calculates more than Euclidean distances as it is based on the routes and the elements influencing the movement and commuting time to reach serum-providing facilities, that is, much closer to the daily reality of the population. Also, unlike the use of points to locate the population, the statistical grid considers the spatial distribution of the population in the territory, which also affects the time-distance relationship. Thus, it contributes to the analysis of access to serum-providing facilities in the country, using a tool that allows integrating several factors that influence travel time and costs.

## Snakebite accidents' information

Besides data used to develop the analysis in ACCESSMOD [16], information about accidents by snake type and municipality, based on place of residence, from 2007 to 2019 was also considered. Data were aggregated by broader spatial scales. This information derives from the

Information System on Notifiable Diseases (SINAN) [6]. The period was selected due to the greater compatibility from the first year of the series than the previous information. Certainly, there are problems and differences about the quality of information about snakebite accidents in SINAN [28, 29], but this health information system has been centralizing the information about notifiable diseases since the beginning of the century [28] and improving its coverage in the last years [30], in all Brazilian States. There is a lack of more recent evaluation about the quality and coverage of the information about snakebite accidents in Brazil.

The results are presented basing on the share population without possibility of reaching a serum providing health facility in two hours for different geographical areas, namely, Brazil, Macro Regions, States, and municipalities, to better characterize the issue.

This analysis focused on access to antiophidic sera, however, it could also be developed to address access to other types of services. The technique allows discussing geographic accessibility under different approaches, considering the location of the facilities and commuting time, and identifying and locating gaps for the care type highlighted here. Therefore, it aims to contribute to the debate on access to health services in the country and points out investment needs to expand the serum supply.

## Results

The analysis of the geographical accessibility to health facilities that can provide antiophidic serum is based on the use of information about Population Distribution, Routes, Land Use, Digital Elevation Model, and Health Facilities. Fig 1 registers the maps of each one of these components.

There is an association between the distribution of population (b), routes (d) and health facilities that can provide serum. More populated parts of the country have more distributed facilities with more expressive availability of routes. The Amazon region, in the northern part of the country, has a spatially concentrated distribution of facilities and more complex circulation or transportation, because in many cases the only available mean are waterways. The Digital Elevation Model (c) was included because ACCESSMOD [16] allows the consideration of effects of going up or down at the roads and the parts of the country which have a mountainous relief. The last map is an example of the Land Use information, just to register its detailing diversity.

The results address how much of the Brazilian population has the possibility to reach serum-providing facilities up to two hours. To this end, several spatial scales were considered to identify gaps. On a national scale, 5% of the Brazilian population could not reach an unit within two hours, which in absolute terms represented about 9.8 million people. Fig 2 shows the distribution of these areas in the country.

In all capital cities and their immediate surroundings, the population could reach a facility within two commuting hours. Concerning inland regions of the states, those with the most significant territorial extensions had the most extensive voids, which was the case of almost all states in the North region, Mato Grosso, Piauí, and Ceará.

The reaching patterns are different between the main regions and, consequently, between the states and the Federal District. The North has the highest proportion of the population (11.8%) more than two hours away, which in absolute terms represents 1.8 million people. In the Northeast, about 5.9 million people (11.4% of the population) are more than two hours away from a facility for antiophidic treatment. In the Midwest, 6.4% of the population are in this situation (885 thousand people), while in the South, this percentage drops to 3.5% (926 thousand people). Finally, in the Southeast this situation is registered only by 0.3% (267 thousand people).

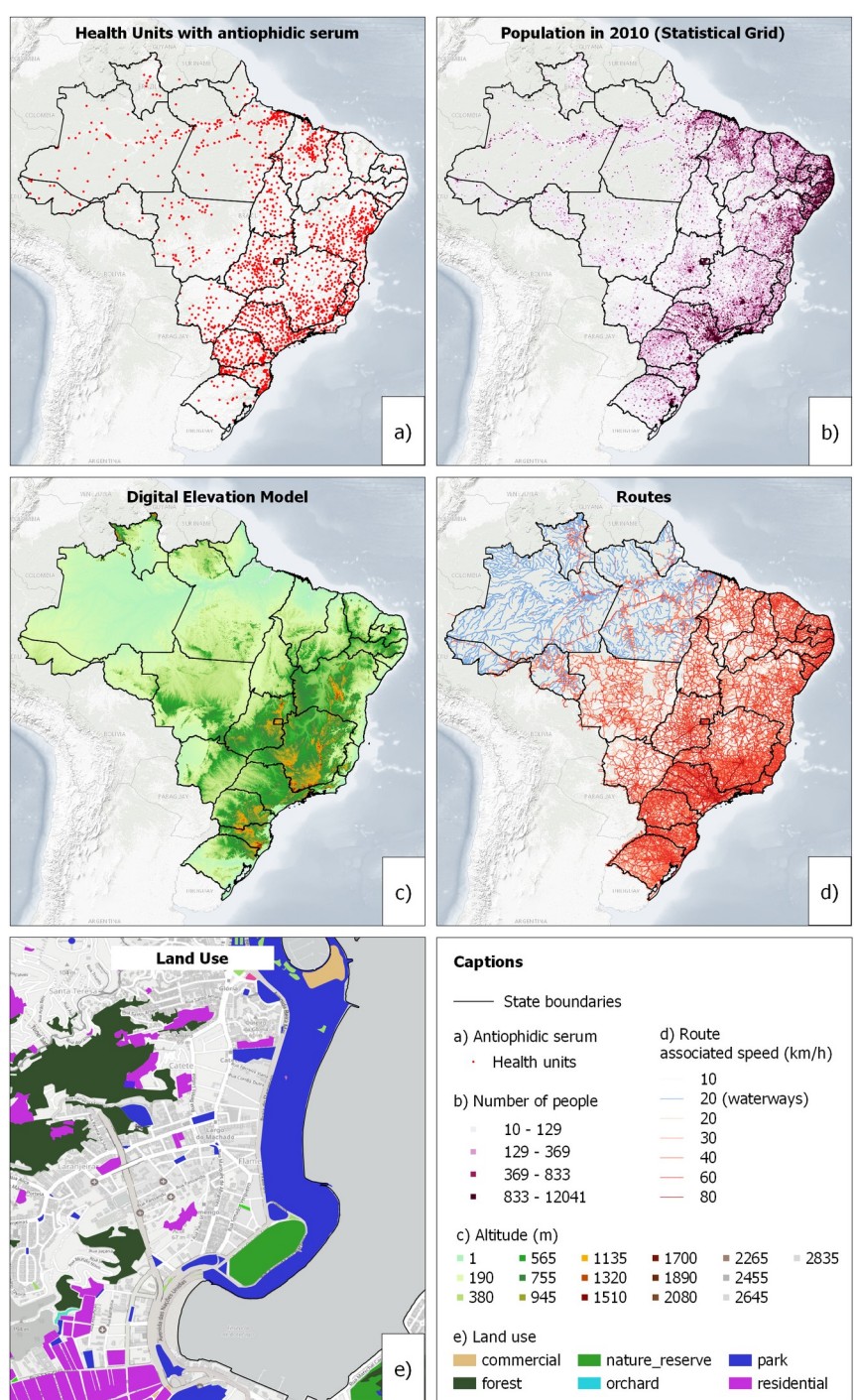

**Fig 1.** Health Facilities (a), Statistical Grid (b), Digital Elevation Model (c), Routes (d) and example of Land Use (e). **Source:** Health Facilities: Toxic-Pharmacological Information System (SINITOX), Statistical Grid: IBGE, Digital Elevation Model: NASA, Background: U.S. Geological Survey/National Geospatial Program, and Transportation and Land Use Routes: Open Street Map.

Considering the States (Fig 3), Maranhão, with 29%, which represents 1.83 million people, has the most significant proportion of the population with more than 2 hours to reach an antiophidic care health unit. Also, the state of Rondônia with 28% of its population (424 thousand

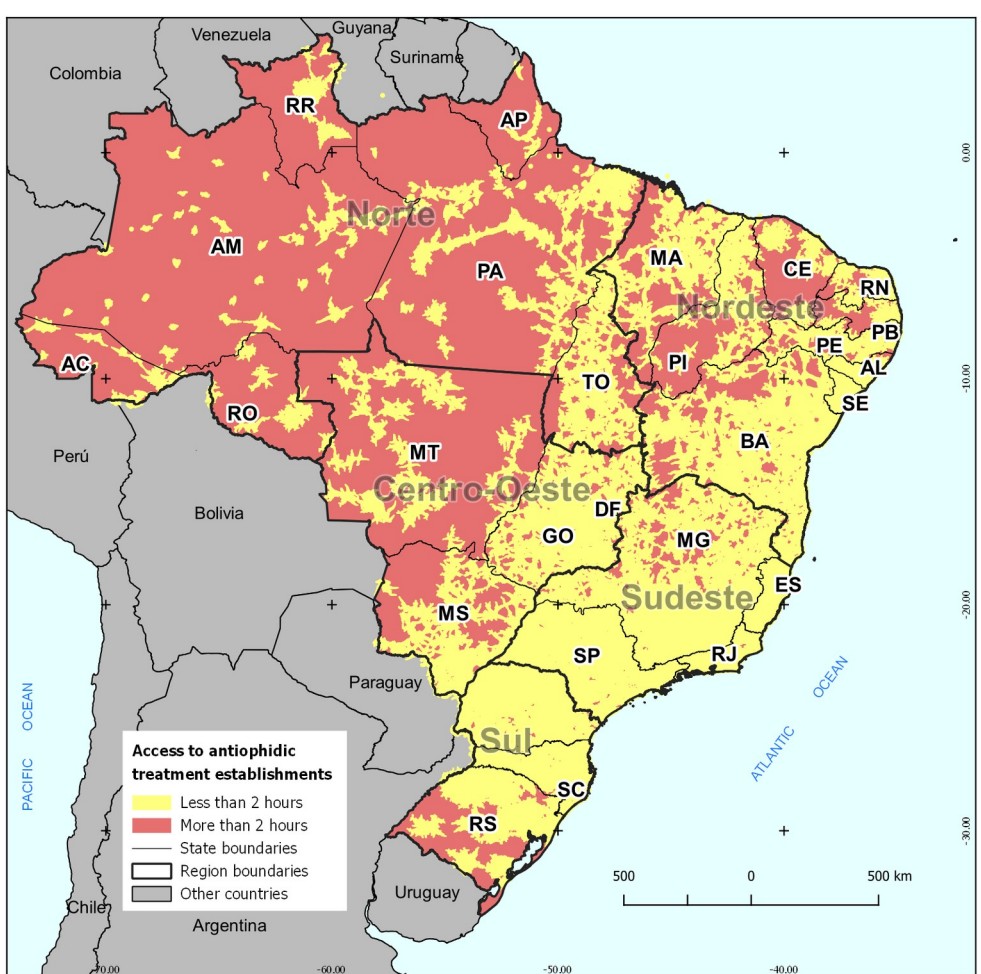

**Fig 2. Population reach to antiophidic treatment by commuting time, Brazil. Source**: IBGE; ICICT/Fiocruz. Organization: Maurício Silva and Diego Ricardo Xavier.

people) and Mato Grosso with 26% (775 thousand people) stand out. In the state of Ceará, 22% of the population stands more than two hours away from antiophidic treatment. However, this represents 1.82 million people, a number similar to Maranhão. When considering only absolute numbers, also relevant are the values observed in the states of Rio Grande do Sul, with 895 thousand people (8.6%), Pará, with 730 thousand people (10.2%), and Paraíba, with 694 thousand people (18.9%), more than 2 hours away from a health unit with antiophidic care.

The last spatial unit used to present the results are municipalities (Fig 4). In this scale, 354 municipalities registered more than 75% of the population without coverage for antiophidic care within a maximum of two commuting hours. Forty-four municipalities stood between 50% and 75% and 150 in the class between 30% and 50%. Four hundred and four municipalities had good coverage, from 15% to 30%, and 4,617 municipalities had less than 15% of uncovered population. Noteworthy are inland areas of the states of Ceará, Paraíba, Piauí, and Rio Grande do Norte, the central region of the state of Mato Grosso and Rondônia, west of Maranhão and Roraima, west of Amazonas, and the extreme South of Rio Grande do Sul.

For comparative purposes, Fig 5 shows the number of snakebite accidents by Brazilian municipality from 2007 to 2019. Except for significant population concentrations such as São

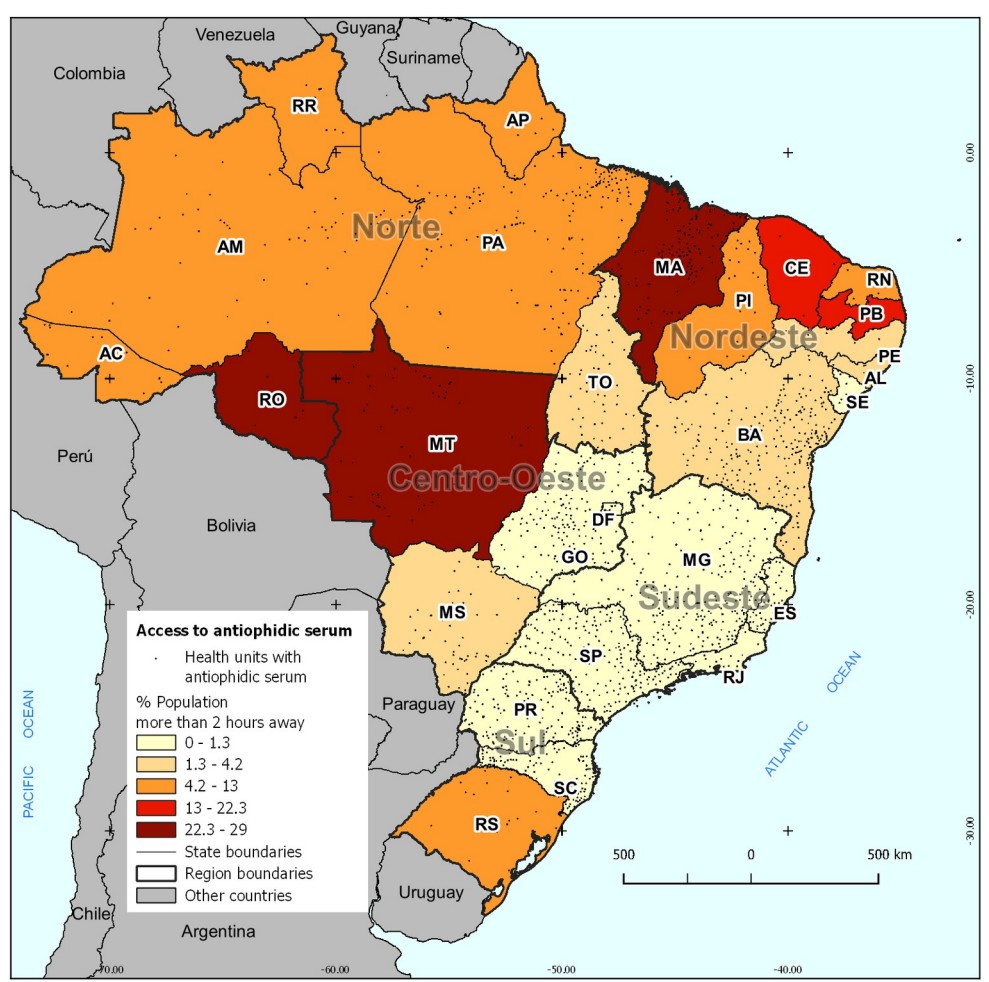

**Fig 3. Population (%) more than 2 hours away from antiophidic treatment, by Brazilian states. Note**: Own elaboration. **Sources**: IBGE; ICICT/FIOCRUZ.

Paulo (SP), Salvador (BA), and the Federal District (DF), which also can register better conditions of notification, most of the municipalities with many accidents are found in the North of the country. Both capitals, such as Manaus (AM), Belém (PA), Rio Branco (AC), Porto Velho (RO), and Macapá (AP), and inland cities are noteworthy, mainly in Pará. It is interesting to highlight a relevant concentration of municipalities with high amounts of accidents in the Lower Amazon basin and along its course in western Pará and eastern Amazon, besides the Solimões River and Upper Negro River, in the Western Amazon.

Based on information from SINAN [6] about snakebite accidents in 2019 and the municipal population estimated by IBGE for 2019 [31], rates were calculated per thousand inhabitants per municipality and are available in a S1 Appendix, that also brings the share of population with accessibility in more than two hours. The Federal District (DF), São Paulo (SP) and Salvador (BA) registered rates below 0.05 accidents per thousand inhabitants, while small municipalities in the Amazon and the Northeast registered values above 3 per thousand. Worth highlighting places like Alto Alegre (6.38) and Uiramutã (5.87) in Roraima, Severiano Melo in Rio Grande do Norte (4.92), Mazagão (4.21) and Itaubal (3.09) in Amapá, Recursolândia in Tocantins (3.96), Afuá in Pará (3.37), and Arame in Maranhão (3.03).

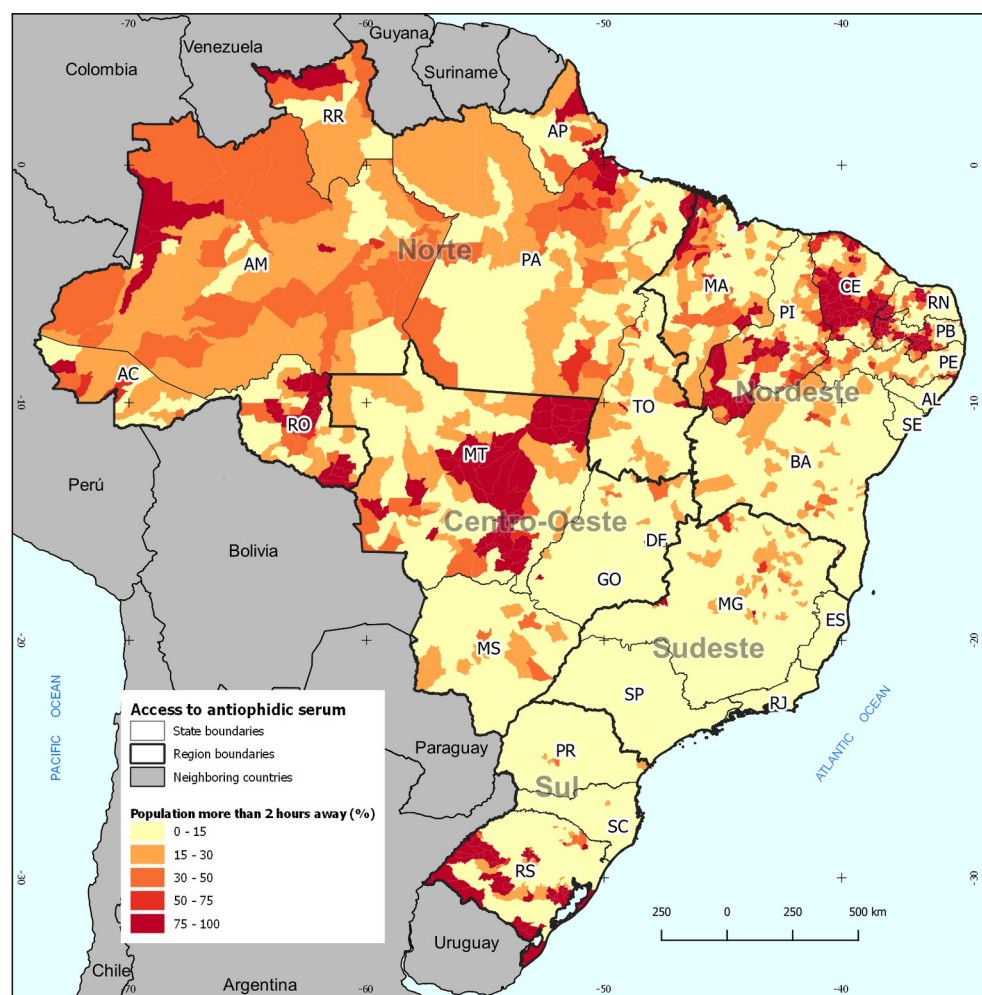

**Fig 4. Population (%) more than 2 hours away from antiophidic treatment, by Brazilian municipalities. Note**: Own elaboration. **Sources**: IBGE; ICICT/FIOCRUZ.

Forty-six municipalities registered incidence rates of 2 or more snakebite accidents in 2019, generally with rural activity patterns [32]. Mostly of them (29) are located in the North Region, characterized by the Amazon Forest, 13 in the state of Pará. The worst condition in terms of population share with more than hours to reach a health facility that provides serum, is registered by northern municipalities, where 13 has more than one third of the population with access difficulties, highlighting four of them with more than 90% in this situation: Anajás (PA), Cutias and Itaubal (AP) and Japurá (AM).

Municipalities with expressive incidence rates but located in other regions of Brazil registered better conditions of geographical accessibility, with shares inferior to 13% of population more than hours away. Only Severiano Melo (RN), a northeastern municipality, had a more relevant participation of this situation (28.9%).

When comparing Figs 4 and 5, combining the most significant difficulties in accessing facilities to the most considerable amounts of snakebites from 2007 to 2019, we observe that the greatest challenges in obtaining serum are found in the Lower Amazon, between the states of Pará and Amapá and west of Manaus (AM), along the Solimões River. The articulation between environmental conditions that lead to a more significant presence of snakes and

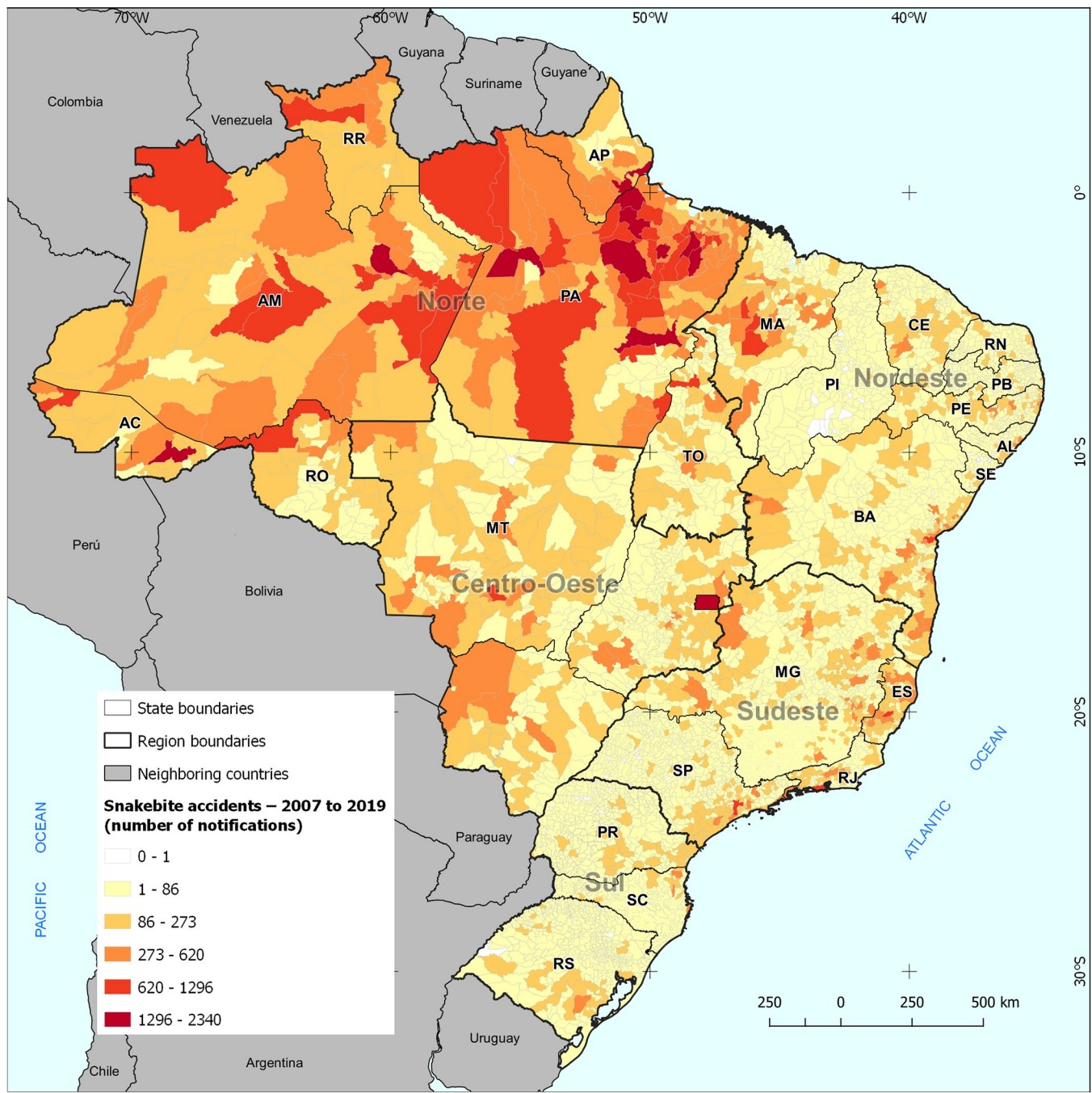

**Fig 5. Number of snakebite accidents from 2007 to 2019 in Brazilian municipalities. Note**: Own elaboration. **Source**: SINAN/DATASUS.

commuting hardships encumber the supply demand. Thus, it is necessary to expand serum supply sites in these regions with a logic that considers accessibility by the means of transportation available in the respective locations.

## Discussion

In Brazil, geographic accessibility is a very important issue due the continental dimension of the country and the existence of vast, remote areas with isolated populations and essential

challenges for people's commuting. In this context, the North stands out and has the highest number of snake accidents and the most significant travel challenge, although it is the least populous region in the country.

The national outlook of accessibility to antiophidic sera reveals a high population coverage, allowing geographic access on time to health facilities that offer this service, what is related to the possibility of reducing deaths and sequelae from these accidents.

This network of serum-providing health facilities is been implemented through years of investment [4], what significantly changed the situation in the supply of sera in the country compared to the one addressed for the state of São Paulo just over three decades ago [33], in combination with a stable national production of antiophidic serum at public institutions [28, 34]. The expanded serum supply network correlates with government investment and the results achieved in the expected outcome, which is case recovery [34]. These studies consider that time between accident and treatment is preponderant for a better prognosis of the patient [13, 29]. While the treatment may have good results two hours after the snakebite accident, a higher success rate can be achieved within this interval.

The segment of the population residing more than two hours away deserves attention since it exceeds 9 million inhabitants. The coverage patterns found in this study reveal that people with the most challenging access live in rural areas or with a predominance of native vegetation. Despite the continental size of Brazil vis-à-vis Costa Rica, the challenges for geographic accessibility to antiophidic sera are similar [13], according to an analysis that also used geoprocessing tools. People residing in remote areas with complex conditions of commuting and transportation (mountainous regions, tropical forests, other sparsely populated areas) characterize complex situations concerning timely access to health care.

Some regions of the country require efforts to expand the offer of antiophidic sera to the population. The first is located in the Northeast, especially in the inland municipalities of the states. The pattern of rural occupation based on small properties [32] puts a high population contingent without the availability of antiophidic serum, and Ceará and Paraíba are the highest priority for the installation of health units offering this service. Inland Piauí and Maranhão can also be mentioned in the same situation.

The Rio Grande do Sul situation is also worrying, even considering the low population density. However, due to the density of the urban network and the circulation routes, the transportation of the injured to the serum-providing sites becomes relatively more manageable since it would be sufficient to direct serum doses to the existing health units.

Another particular reality is that of northern Mato Grosso, where the sizeable rural property predominates [32]. It is a low population density area where municipalities have large extensions. Thus, focusing efforts only on making serum available in small urban centers may not be enough, and the articulation between government and farm owners is an additional option for timely delivery of antiophidic serum to rural workers in the region.

Finally, the expressive territorial extension of the Amazon, especially the inland region of the states of Acre, Amazonas, Rondônia, Roraima, Amapá, and eastern Pará, is characterized by an extensively dispersed population, with forests, and waterway commuting. Alternative actions should be considered in these areas, focusing on those based on the availability of antiophidic serum closer to the population and expanding the network of health facilities. Also, actions that implement an efficient distance integration (from the accident site to the nearest health unit) and agile rescue means that use speedboats or helicopters can become a more appropriate option in the Amazon reality.

In more than 100 years, studies on this theme show that the epidemiological profile of snakebite cases has not changed [4]. The cases occur more frequently at the beginning and the end of the year, in male rural workers aged 15–49 years, mainly affecting the lower limbs.

Without an individual consideration of risk, the characterization of areas in the present study reveals situations of important incidence and low geographical accessibility, with concentration of rural activities, especially in the North, but also at the Northeast region and the state of Mato Grosso.

It is important to consider that the location of services hinders access to health services concerning supply, while the means of transportation available and commuting costs are related to demand [10, 11]. The travel time for obtaining antiophidic sera is related to all identified barriers and expresses critical challenges for obtaining resolutive care. All those questions are related to geographical accessibility, but access to health services is a much broader question [7–9] and in the case of antiophidic sera involves at least the actual availability of serum, which depends on production and distribution logistics.

The study has some limitations apart from not considering the availability of serum, which would depend on qualified information on antiophidic sera distribution. The postponement of Brazilian 2020 Demographic Census imposed the need of using population data of the last Census, in 2010. The difficulties to estimate the travel speed considering rivers and other waterways (a widespread situation in the Amazon Region) also brings difficulties to the present approach. Also, the fact that there is an interval between the snakebite accident and the access to transportation, that can imply in a significant preliminary delay is very relevant, with impacts in the two-hour time to reach treatment, but is impossible to infer in a national perspective.

Some Brazilian studies in Brazil have also proposed to assess the time from snakebite to patient care [14, 15]. However, these are local studies starting from a casualty base, and their various outcomes are observed, considering specific small-scale analysis units. The innovation of this study is observing on a national scale the population at a distance that, according to the literature, brings more significant harm to treatment. In this sense, care gaps that can direct interventions and, consequently, reduce the cases developing to death or significant sequelae to the patient are pointed out.

## Conclusions

This study contributes with the analysis of geographical accessibility integrating several issues that influences commuting time and this can be developed for other healthcare networks. Other contributions are related to the analysis of access barriers linked to travel hardships, which imply several costs, and identification of places with investment needs to expand antiophidic sera offer, either by installing more health units or improving the injured transport conditions. In this perspective, the study can subsidize other studies about geographical accessibility and the actual academic and public policies debate about the questions that are dealt. Also, it is important to highlight that the analysis also shows the relevance of well-structured and updated Health Information Systems for planning health actions and their potential use.

## Supporting information

**S1 Appendix. Snakebite accidents, population, incidence rate and uncovered population (%) by Brazilian municipalities, 2019.**
(DOCX)

## Author Contributions

**Conceptualization:** Ricardo Antunes Dantas de Oliveira, Diego Ricardo Xavier Silva.

**Data curation:** Diego Ricardo Xavier Silva, Maurício Gonçalves e Silva.

**Formal analysis:** Diego Ricardo Xavier Silva, Maurício Gonçalves e Silva.

**Investigation:** Diego Ricardo Xavier Silva.

**Methodology:** Ricardo Antunes Dantas de Oliveira, Maurício Gonçalves e Silva.

**Project administration:** Ricardo Antunes Dantas de Oliveira.

**Software:** Diego Ricardo Xavier Silva, Maurício Gonçalves e Silva.

**Supervision:** Ricardo Antunes Dantas de Oliveira.

**Validation:** Ricardo Antunes Dantas de Oliveira, Maurício Gonçalves e Silva.

**Visualization:** Ricardo Antunes Dantas de Oliveira, Diego Ricardo Xavier Silva, Maurício Gonçalves e Silva.

**Writing – original draft:** Ricardo Antunes Dantas de Oliveira, Diego Ricardo Xavier Silva, Maurício Gonçalves e Silva.

**Writing – review & editing:** Ricardo Antunes Dantas de Oliveira, Diego Ricardo Xavier Silva.

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
