## [Decision Letter · Decision Letter 0]

26 Jul 2021

PONE-D-21-22192

Geographical accessibility to the supply of antiophidic sera in Brazil: Timely access possibilities

PLOS ONE

Dear Dr. Oliveira,

Thank you for submitting your manuscript to PLOS ONE. After careful consideration, we feel that it has merit but does not fully meet PLOS ONE’s publication criteria as it currently stands. Therefore, we invite you to submit a revised version of the manuscript that addresses the points raised during the review process.

Please consider all comments

We look forward to receiving your revised manuscript.

Kind regards,

Ahmed Mancy Mosa, Ph.D.

Academic Editor

PLOS ONE

Journal Requirements:

"we acknowledge the Inova Fiocruz Program for the funding."

"RAD Oliveira have funding from Inova Fiocruz Program (https://portal.fiocruz.br/programa-inova-fiocruz), a research support obtained in a selection occurred in 2018. The funds allowed the payment of scholarships to organize the datasets and the spatial information treatment, beyond the participation in congresses and seminars. The funding body had no role in the study design, data collection and analysis, decision to publish, or preparation of the manuscript."

5. We note that Figures 1 to 5 in your submission contain [map/satellite] images which may be copyrighted. All PLOS content is published under the Creative Commons Attribution License (CC BY 4.0), which means that the manuscript, images, and Supporting Information files will be freely available online, and any third party is permitted to access, download, copy, distribute, and use these materials in any way, even commercially, with proper attribution. For these reasons, we cannot publish previously copyrighted maps or satellite images created using proprietary data, such as Google software (Google Maps, Street View, and Earth). For more information, see our copyright guidelines: http://journals.plos.org/plosone/s/licenses-and-copyright.

a. You may seek permission from the original copyright holder of Figures 1 to 5

to publish the content specifically under the CC BY 4.0 license.  

Reviewers' comments:

Reviewer's Responses to Questions

Comments to the Author

1. Is the manuscript technically sound, and do the data support the conclusions?

Reviewer #1: Yes

Reviewer #2: Partly

Reviewer #3: Yes

2. Has the statistical analysis been performed appropriately and rigorously?

Reviewer #1: Yes

Reviewer #2: N/A

Reviewer #3: Yes

3. Have the authors made all data underlying the findings in their manuscript fully available?

Reviewer #1: Yes

Reviewer #2: Yes

Reviewer #3: Yes

4. Is the manuscript presented in an intelligible fashion and written in standard English?

Reviewer #1: Yes

Reviewer #2: Yes

Reviewer #3: Yes

5. Review Comments to the Author

Reviewer #1: Abstract

- Background must clearly describe the purpose of the study.

- Methods must describe: the type of study design and the method used (ACCESSMOD), the data collection and analysis, and the measures used in the description of Results, summarizing what was presented in the Methods section of the manuscript.

- In Results, the states that present the longest time to health units should be highlighted.

Background

- The paragraphs in lines 106-113 and 127-138 would be more suitable in the Methods section and not in the Background.

- The last paragraph of the Background should describe the purpose of the study.

Methods

- The type of study design must be described in the first paragraph.

- After the subtitle and paragraph referring to “Distance calculation”, it is necessary to describe in detail the tabulation and data analysis process, as well as the measures which were used, statistical tests and the level of significance considered.

- The authors used 3 distinct geographic analysis units: macro-regions, federative units (states and Federal District) and municipalities. This should also be described in Methods.

Results

- In lines 277-285, I suggest that the calculation of the incidence rate be revised, since the indicator in the DF, São Paulo and Salvador has been presented as 0.05. It would be more appropriate to work with the indicator per 100,000 inhabitants.

Discussion

It is important to discuss not only the innovation proposed by this study, but also its limitations, at the end of the section.

Reviewer #2: Snakebite accidents are, in some regions, relevant public health issues. The WHO's recognition, including this problem in the list of neglected tropical diseases, was an important step towards the production of more knowledge on the subject and, mainly, the mobilization of governments and other institutions to reduce the damage caused by this type of event. The manuscript has the merit of addressing this topic, which is of regional relevance in Brazil.

I understand that the central focus of the work and its strength are in the application of the method that analyzes the distance between health units and population distribution. As the author himself mentions, it could be used for other health situations. The problem is using this single analysis to base the entire discussion of the manuscript.The subject of snake accidents ends up being secondary and treated in a superficial way or weakly based on the results obtained from the performed analyses. Fundamental factors were left out or were superficially addressed, if we consider the endo f the title “Timely access possibilities.” For example, the availability of serum.

The manuscript ignores the limited availability of the serum and, therefore, assumes the existence of an unlimited amount of the product. It assumes that adequate allocation in health units would only lack evidence-based rationalization (the model presented, for example). Unfortunately, that's not the reality. The distribution of sera is fundamentally conditioned by its availability.

The WHO classification itself highlights this discussion. It is not just the affected population that defines an illness as neglected. The industry's insufficient investment as well. In the case of the production of serums, this is even more sensitive, since the production, usually, has to be regionalized according to the specificity of the snake species, with specific distribution.

Not every population more than two hours away from the health units that provide serum is under the same risk of snakebiting. The categorization of the most isolated populations in terms of risk would be essential to better qualify the analysis and enable a prioritization criterion to support decision-making and suggest “timely access possibilities” as the title says. Municipalities with higher incidences are briefly mentioned, but the discussion is not go further.

Furthermore, the quality of the information is not discussed. For example, when commenting on the large number of cases in the city of São Paulo, the author mentions only the large population of the city as an explanation and does not assess the possible effect on the notifications of the existence of very important references (regional and national) in the treatment of snakebite accidents, located in its territory.

For these reasons, it seems a little ambitious to discuss "timely access possibilities", based almost exclusively on the distance between the population and the health unit, without analyzing risk, serum availability, health system configuration, information quality.

There are some problems with the references. For example, information on the WHO classification of snakebites as a neglected tropical disease is referenced to the article "Longbottom J, Shearer FM, Devine M, Alcoba G, Chappuis F, Weiss DJ, Williams, DJ. Vulnerability to snakebite envenoming: A global mapping of hotspots. The Lancet. 2018; 392(10148): 673-84". However, the information is not from this article, which cites original information from WHO document "Report of the tenth meeting of the WHO Strategic and Technical Advisory Group for neglected tropical diseases. World Health Organization, Geneva2017".

In my opinion, the work presented allows discussing only the geographic distribution of the population in relation to the units that provide serum, without assuming that this analysis is sufficient to think about strategies for expanding access to services.

Reviewer #3: This manuscript assessed the accessibility to antiophidic sera in the Brazilian context. Snakebite accidents are a neglected public health problem in the country; the knowledge aimed at by the present study is certainly relevant to health policy and planning in many remote and rural areas. The text is well written and can be read smoothly. However, it is poorly organized.

The Introduction is unnecessarily too large. It should have ended in line 97, after stating the study objective. Using different geographical areas (lines 97-8) in the approach is related to Methods, not the Introduction. That this analysis allows discussing accessibility to other health services (lines 99-105) is a matter for Discussion. The remaining four paragraphs (lines 106-138) pertain to Methods, an essential part of Methods.

Figure 1 (lines 142, 163-5) should be replaced in Results. Just describing which were the variables collated in this study is enough for this segment of Methods. Depicting graphically the Results is a strength of the study. The maps provide a synthetical perception of the problem by region, state, and town. However, I missed a tabular display of cities and towns with higher or lower accessibility, maybe as a supplement. Such information would be critical to document the problem and guide solution attempts at the municipal level.

The figures have poor visual accuracy; they need a higher resolution. They also need a more explicative heading. Figure 1’s heading is meaningless. Figures 2-4’s headings are relatively interchangeable; I missed an unequivocal indication of which figure is showing accessibility by which criteria.

I observe that the North region had a higher number of snake accidents (Figure 5) and the most significant travel challenge (Figures 2-4). This remark is the study’s conclusion; it could not have been preliminarily stated in the Introduction (lines 92-4) without referring this information to the literature or to empirical data (which, of course, were subsequently reported).

I missed the acknowledgment of study limitations. The Discussion’s last paragraph pointed out the strength of having gathered an extensive, nationwide database. However, traveling to the health unit that offers sera is not the only interval to the treatment. In particular, the period from the casualty and the access to transport may result in a significant preliminary delay. This study cannot infer this matter, though this delay can enlarge the time to treatment. I suggest that the authors discuss this issue as a study limitation.

6. PLOS authors have the option to publish the peer review history of their article (what does this mean?). If published, this will include your full peer review and any attached files.

Do you want your identity to be public for this peer review? For information about this choice, including consent withdrawal, please see our Privacy Policy.

Reviewer #1: No

Reviewer #2: No

Reviewer #3: Yes: Jose Leopoldo Ferreira Antunes

---

## [Author Response · Author response to Decision Letter 0]

12 Aug 2021

Rio de Janeiro, August 12th, 2021

Rebuttal Letter 

Dear editors and reviewers,

Thank you for the revision and suggestions to improve our manuscript. In order to resubmit it, we answer each one of the points that the academic editor and the reviewers raised about our work. First, we respond to the Journal Requirements and then to all the requests and suggestions, putting the points in italic and our answers just below.

Journal Requirements

All PLOS ONE’s style requirements were reviewed and corrected when the manuscript had something distinct from it. The front page, headings and subheadings’ size, citation format and figures indications were the main corrections done.

I have provided the correct grant number and match Financial Information and Financial Disclosure when I resubmitted it.

"we acknowledge the Inova Fiocruz Program for the funding."

"RAD Oliveira have funding from Inova Fiocruz Program (https://portal.fiocruz.br/programa-inova-fiocruz), a research support obtained in a selection occurred in 2018. The funds allowed the payment of scholarships to organize the datasets and the spatial information treatment, beyond the participation in congresses and seminars. The funding body had no role in the study design, data collection and analysis, decision to publish, or preparation of the manuscript."

I have provided funding information in our Funding Statement and included it in our cover letter. I removed financial information from the manuscript. Please change my funding statement to:

RAD Oliveira have funding from Inova Fiocruz Program (https://portal.fiocruz.br/programa-inova-fiocruz), (Grant number: Inova Program VPPCB-008-FIO-18-2-48), a research support obtained in a selection occurred in 2018. The funds allowed the payment of scholarships to organize the datasets and the spatial information treatment, beyond the participation in congresses and seminars. The funding body had no role in the study design, data collection and analysis, decision to publish, or preparation of the manuscript.

In order to cope with PLOS ONE’s requirements, we informed in our cover letter that there are no restrictions over the study’s data availability. The data we generated can be found at: https://doi.org/10.7303/syn26042133. The data we have obtained from other sources are detailed in our submission.

5. We note that Figures 1 to 5 in your submission contain [map/satellite] images which may be copyrighted. All PLOS content is published under the Creative Commons Attribution License (CC BY 4.0), which means that the manuscript, images, and Supporting Information files will be freely available online, and any third party is permitted to access, download, copy, distribute, and use these materials in any way, even commercially, with proper attribution. For these reasons, we cannot publish previously copyrighted maps or satellite images created using proprietary data, such as Google software (Google Maps, Street View, and Earth). For more information, see our copyright guidelines: http://journals.plos.org/plosone/s/licenses-and-copyright.

a. You may seek permission from the original copyright holder of Figures 1 to 5

to publish the content specifically under the CC BY 4.0 license. 

In order to solve the demand about copyrighted figures we decided to change the background of Figure 1 using the suggested USGS National Map Viewer (public domain): http://viewer.nationalmap.gov/viewer/. That figure was only on with Google Maps background. The other figures were built using Brazilian Institute of Geography and Statistics (IBGE – https://www.ibge.gov.br/apps/basescartograficas/) cartographic base, which have public domain and are available to reuse considering the metadata that follows ISO 19115 as standard. The figure captions were updated with this information.

Review Comments to the Author

Reviewer #1: Abstract

- Background must clearly describe the purpose of the study.

- Methods must describe: the type of study design and the method used (ACCESSMOD), the data collection and analysis, and the measures used in the description of Results, summarizing what was presented in the Methods section of the manuscript.

- In Results, the states that present the longest time to health units should be highlighted.

The abstract was updated considering the requests and suggestions, so the purpose of the study was included, the methods were more detailed and the States with more problems of accessibility were highlighted. 

Background

- The paragraphs in lines 106-113 and 127-138 would be more suitable in the Methods section and not in the Background.

- The last paragraph of the Background should describe the purpose of the study.

The section was changed considering the need of reallocation of some paragraphs to Methods and the purpose of the study defines the last paragraph of the Introduction.

Methods

- The type of study design must be described in the first paragraph.

- After the subtitle and paragraph referring to “Distance calculation”, it is necessary to describe in detail the tabulation and data analysis process, as well as the measures which were used, statistical tests and the level of significance considered.

- The authors used 3 distinct geographic analysis units: macro-regions, federative units (states and Federal District) and municipalities. This should also be described in Methods.

The type of the study was included in the paragraph of the Methods section, which also had the inclusion of the three distinct geographic analysis units. The Distance Calculation subsection was updated with more information about the calculation process developed on ACCESSMOD, which considers Dijkstra least-cost path algorithm. This last information make reference to an article that presents the software:

Ray N, Ebener S. AccessMod 3.0: computing geographic coverage and accessibility to health care services using anisotropic movement of patients. Int J Health Geogr. 2008; https://doi.org/10.1186/1476-072X-7-63

 Results

- In lines 277-285, I suggest that the calculation of the incidence rate be revised, since the indicator in the DF, São Paulo and Salvador has been presented as 0.05. It would be more appropriate to work with the indicator per 100,000 inhabitants.

The vast majority of Brazilian municipalities has less than 100,000 inhabitants, so we prefer to calculate the incidence rate per 1,000 to keep a more realistic comparison. According to the 2019 Population Estimates, presented by IBGE, 44% of the municipalities had less than 10,000 inhabitants and 94.2% had less than 100,000.

Discussion

It is important to discuss not only the innovation proposed by this study, but also its limitations, at the end of the section.

The limitations of the study were updated with more details and discussions. It is located at the penultimate paragraph of the Section.

Reviewer #2: 

Snakebite accidents are, in some regions, relevant public health issues. The WHO's recognition, including this problem in the list of neglected tropical diseases, was an important step towards the production of more knowledge on the subject and, mainly, the mobilization of governments and other institutions to reduce the damage caused by this type of event. The manuscript has the merit of addressing this topic, which is of regional relevance in Brazil.

I understand that the central focus of the work and its strength are in the application of the method that analyzes the distance between health units and population distribution. As the author himself mentions, it could be used for other health situations. The problem is using this single analysis to base the entire discussion of the manuscript.The subject of snake accidents ends up being secondary and treated in a superficial way or weakly based on the results obtained from the performed analyses. Fundamental factors were left out or were superficially addressed, if we consider the endo f the title “Timely access possibilities.” For example, the availability of serum.

The manuscript ignores the limited availability of the serum and, therefore, assumes the existence of an unlimited amount of the product. It assumes that adequate allocation in health units would only lack evidence-based rationalization (the model presented, for example). Unfortunately, that's not the reality. The distribution of sera is fundamentally conditioned by its availability.

The WHO classification itself highlights this discussion. It is not just the affected population that defines an illness as neglected. The industry's insufficient investment as well. In the case of the production of serums, this is even more sensitive, since the production, usually, has to be regionalized according to the specificity of the snake species, with specific distribution.

To cope with these comments we try to amplify the characterization of the study as an analysis of geographical accessibility, not access or even use in fact of the services. To do so, a more theoretical paragraph about the complex concept of access and its dimensions, which includes geographical accessibility, was written in the Introduction. In addition, we have increased the limitation approach in the Discussion section in order to make clear that the actual availability of serum is not a subject of the manuscript, but has fundamental importance in the question.

The definition of snakebite accidents as neglected disease subsidizes our evaluation of the difficulties to access health facilities in terms of the location and the costs due the need of commuting, especially in regions with remote population and poor availability of transportation means and routes. 

In Brazil, three public funded research institutions also produce antiophidic sera: Instituto Butantan (https://butantan.gov.br/), Fundação Ezequiel Dias (http://www.funed.mg.gov.br/) e Instituto Vital Brazil (http://www.vitalbrazil.rj.gov.br/index.html). Many authors consider that the production in our country is stable since the beginning of the 21st century as we include in the Discussion section. Obviously, this does not mean that there are no problems in production or distribution, but allows to a better situation when comparing to other developing countries.

Not every population more than two hours away from the health units that provide serum is under the same risk of snakebiting. The categorization of the most isolated populations in terms of risk would be essential to better qualify the analysis and enable a prioritization criterion to support decision-making and suggest “timely access possibilities” as the title says. Municipalities with higher incidences are briefly mentioned, but the discussion is not go further.

To deal with this comment we include more paragraphs at Results and Discussion’ sections. The characterization of risk was done considering geographical units with important incidence in 2019 and poor geographical accessibility to health facilities. We develop more paragraphs about the worst situations between Brazilian municipalities and include a Supplemental File with the municipalities with incidence rate in 2019 and population not covered in two-hour time. Our approach is related to the possibility of reach a facility that could provide antiophidic serum, hoping that it can be useful in planning process to expand the access to this network. Therefore, it is the possible access in opportune time that we analyse.

Furthermore, the quality of the information is not discussed. For example, when commenting on the large number of cases in the city of São Paulo, the author mentions only the large population of the city as an explanation and does not assess the possible effect on the notifications of the existence of very important references (regional and national) in the treatment of snakebite accidents, located in its territory.

We include comments and references about the quality of information about snakebite accidents in a new subsection created in Methods. The information we use refers to place of residence, so the effect of health care references is reduced, but for sure there is a possibility of better notification in some regions and we pointed it in the text. There is a lack of more recent studies about the quality of the information about snakebite accidents on SINAN.

For these reasons, it seems a little ambitious to discuss "timely access possibilities", based almost exclusively on the distance between the population and the health unit, without analyzing risk, serum availability, health system configuration, information quality.

We consider the timely access possibilities from the perspective of geographical accessibility to health facilities. It does not mean that we consider our approach as sufficient to deal with all the complexities of the question, we understand its limitations, but we think even so is a relevant contribution to public policies in this subject. 

There are some problems with the references. For example, information on the WHO classification of snakebites as a neglected tropical disease is referenced to the article "Longbottom J, Shearer FM, Devine M, Alcoba G, Chappuis F, Weiss DJ, Williams, DJ. Vulnerability to snakebite envenoming: A global mapping of hotspots. The Lancet. 2018; 392(10148): 673-84". However, the information is not from this article, which cites original information from WHO document "Report of the tenth meeting of the WHO Strategic and Technical Advisory Group for neglected tropical diseases. World Health Organization, Geneva2017".

The WHO report was included in the References of the manuscript.

In my opinion, the work presented allows discussing only the geographic distribution of the population in relation to the units that provide serum, without assuming that this analysis is sufficient to think about strategies for expanding access to services.

We do not think that this is sufficient to think about strategies to expand access to health services, but we think it as a contribution to the academic debate and to the planning process.

Reviewer #3: 

This manuscript assessed the accessibility to antiophidic sera in the Brazilian context. Snakebite accidents are a neglected public health problem in the country; the knowledge aimed at by the present study is certainly relevant to health policy and planning in many remote and rural areas. The text is well written and can be read smoothly. However, it is poorly organized.

The Introduction is unnecessarily too large. It should have ended in line 97, after stating the study objective. Using different geographical areas (lines 97-8) in the approach is related to Methods, not the Introduction. That this analysis allows discussing accessibility to other health services (lines 99-105) is a matter for Discussion. The remaining four paragraphs (lines 106-138) pertain to Methods, an essential part of Methods.

In order to deal with the perception of poor organization, we made some changes in the manuscript. The Introduction was reduced with the transference of parts to Methods and to the Discussion section, just as suggested. 

Figure 1 (lines 142, 163-5) should be replaced in Results. Just describing which were the variables collated in this study is enough for this segment of Methods. 

Agreeing with this suggestion, Figure 1 was replaced in Results, with the inclusion of a paragraph where some comments about it were included.

Depicting graphically the Results is a strength of the study. The maps provide a synthetical perception of the problem by region, state, and town. However, I missed a tabular display of cities and towns with higher or lower accessibility, maybe as a supplement. Such information would be critical to document the problem and guide solution attempts at the municipal level.

Considering the relevance of such suggestion, a tabular display of the municipalities with incidence rate in 2019 and population not covered in two-hour time was included in the submission as Supplemental File. We believe that can published as .csv or even a text document. Additionally, other comments about the question were included in the Discussion.

The figures have poor visual accuracy; they need a higher resolution. They also need a more explicative heading. Figure 1’s heading is meaningless. Figures 2-4’s headings are relatively interchangeable; I missed an unequivocal indication of which figure is showing accessibility by which criteria.

A higher resolution would increase the file size, so it is not possible to replace them. To cope with demand of improving the headings, we change those from Figures 1, 3 and 4, with intent to clarify its content. Figure 2-4 registers the accessibility of population of defined geographic units that can access serum providing health facilities in two-hour time. However, while Figure 2 register the distribution of the reaching coverage, Figures 3 and 4 registers the share of population with possibility to reach the considered health facilities, by States (Fig 3) and by municipalities (Fig 4). 

I observe that the North region had a higher number of snake accidents (Figure 5) and the most significant travel challenge (Figures 2-4). This remark is the study’s conclusion; it could not have been preliminarily stated in the Introduction (lines 92-4) without referring this information to the literature or to empirical data (which, of course, were subsequently reported).

Agreeing with this suggestion, all the comments about the relevance of the question at the North region were transferred to Discussion section.

I missed the acknowledgment of study limitations. The Discussion’s last paragraph pointed out the strength of having gathered an extensive, nationwide database. However, traveling to the health unit that offers sera is not the only interval to the treatment. In particular, the period from the casualty and the access to transport may result in a significant preliminary delay. This study cannot infer this matter, though this delay can enlarge the time to treatment. I suggest that the authors discuss this issue as a study limitation.

The study limitations were updated with more details and discussion, including the question of the time between the snakebite accident and the actual access to transportation.

Kind regards,

Ricardo Antunes Dantas de Oliveira, Diego Ricardo Xavier Silva and Maurício Gonçalves e Silva

---

## [Decision Letter · Decision Letter 1]

6 Oct 2021

PONE-D-21-22192R1Geographical accessibility to the supply of antiophidic sera in Brazil: Timely access possibilitiesPLOS ONE

Dear Dr. Oliveira,

Thank you for submitting your manuscript to PLOS ONE. After careful consideration, we feel that it has merit but does not fully meet PLOS ONE’s publication criteria as it currently stands. Therefore, we invite you to submit a revised version of the manuscript that addresses the points raised during the review process.

Please consider all comments

We look forward to receiving your revised manuscript.

Kind regards,

Ahmed Mancy Mosa, Ph.D.

Academic Editor

PLOS ONE

Journal Requirements:

Reviewers' comments:

Reviewer's Responses to Questions

**Comments to the Author**

1. If the authors have adequately addressed your comments raised in a previous round of review and you feel that this manuscript is now acceptable for publication, you may indicate that here to bypass the “Comments to the Author” section, enter your conflict of interest statement in the “Confidential to Editor” section, and submit your "Accept" recommendation.

Reviewer #1: All comments have been addressed

Reviewer #2: All comments have been addressed

Reviewer #3: All comments have been addressed

2. Is the manuscript technically sound, and do the data support the conclusions?

Reviewer #1: Yes

Reviewer #2: Yes

Reviewer #3: Yes

3. Has the statistical analysis been performed appropriately and rigorously? 

Reviewer #1: Yes

Reviewer #2: N/A

Reviewer #3: Yes

4. Have the authors made all data underlying the findings in their manuscript fully available?

Reviewer #1: Yes

Reviewer #2: Yes

Reviewer #3: Yes

5. Is the manuscript presented in an intelligible fashion and written in standard English?

Reviewer #1: Yes

Reviewer #2: Yes

Reviewer #3: Yes

6. Review Comments to the Author

Reviewer #1: Abstract

- Methods must describe: the type of study design.

Methods

- Data analysis: it is necessary to describe, as well as the association measures which were used, the statistical tests and the level of significance considered.

Reviewer #2: I'd like to thank all the authors for their answers. The theme is relevant to public health, especially because it affects more seriously socially vulnerable communities. I have no further comments.

Reviewer #3: (No Response)

7. PLOS authors have the option to publish the peer review history of their article (what does this mean?). If published, this will include your full peer review and any attached files.

Reviewer #1: No

Reviewer #2: No

Reviewer #3: **Yes: **Jose Leopoldo Ferreira Antunes

---

## [Author Response · Author response to Decision Letter 1]

22 Oct 2021

Dear editors and reviewers,

Thank you for the revision and suggestions to improve our manuscript. In order to resubmit it, we answer the points that the academic editor and one of the reviewers raised about our work. First, we respond to the Journal Requirements and then to the requests and suggestions, putting the points in italic and our answers just below.

Journal Requirements

The reference list was reviewed in order to correct it, considering the reference style of International Committee of Medical Journal Editors (ICMJE). As a new reference was included, the reference indications were changed in the text and also the reference list.

Review Comments to the Author

Reviewer #1: Abstract

Methods must describe: the type of study design.

The abstract was updated considering the request, adding the type of study design.

Methods

Data analysis: it is necessary to describe, as well as the association measures which were used, the statistical tests and the level of significance considered.

In order to improve the information about the travelling time calculation, the first paragraph of the Methods section was changed. Also, a new reference was included, to better characterize the algorithm considered in ACCESSMOD 5.0, the software which was used to estimate traveling time to health facilities. The Distance and time to reach health facilities calculation subsection was altered in order to include this information.

There is no other information about the association measures which were used, the statistical tests and the level of significance considered, because we do not use specific statistical analysis and even other authors that use the software do not bring those details. But the referred new article can provide a more detailed information about the travel time cost surface model.

Dijkstra, E.W. A note on two problems in connexion with graphs. Numer. Math. 1959;1: 269–271. https://doi.org/10.1007/BF01386390

Kind regards,

Ricardo Antunes Dantas de Oliveira, Diego Ricardo Xavier Silva and Maurício Gonçalves e Silva

---

## [Decision Letter · Decision Letter 2]

8 Nov 2021

Geographical accessibility to the supply of antiophidic sera in Brazil: Timely access possibilities

PONE-D-21-22192R2

Dear Dr. Oliveira,

We’re pleased to inform you that your manuscript has been judged scientifically suitable for publication and will be formally accepted for publication once it meets all outstanding technical requirements.

Kind regards,

Ahmed Mancy Mosa, Ph.D.

Academic Editor

PLOS ONE

Additional Editor Comments (optional):

Reviewers' comments:

Reviewer's Responses to Questions

**Comments to the Author**

1. If the authors have adequately addressed your comments raised in a previous round of review and you feel that this manuscript is now acceptable for publication, you may indicate that here to bypass the “Comments to the Author” section, enter your conflict of interest statement in the “Confidential to Editor” section, and submit your "Accept" recommendation.

Reviewer #1: All comments have been addressed

2. Is the manuscript technically sound, and do the data support the conclusions?

Reviewer #1: Yes

3. Has the statistical analysis been performed appropriately and rigorously? 

Reviewer #1: Yes

4. Have the authors made all data underlying the findings in their manuscript fully available?

Reviewer #1: Yes

5. Is the manuscript presented in an intelligible fashion and written in standard English?

Reviewer #1: Yes

6. Review Comments to the Author

Reviewer #1: (No Response)

7. PLOS authors have the option to publish the peer review history of their article (what does this mean?). If published, this will include your full peer review and any attached files.

Reviewer #1: No

---

## [Editor Report · Acceptance letter]

4 Jan 2022

PONE-D-21-22192R2 

Geographical accessibility to the supply of antiophidic sera in Brazil: Timely access possibilities 

Dear Dr. Oliveira:

I'm pleased to inform you that your manuscript has been deemed suitable for publication in PLOS ONE. Congratulations! Your manuscript is now with our production department. 

Kind regards, 

on behalf of

Dr. Ahmed Mancy Mosa 

Academic Editor

PLOS ONE